# A Qualitative Comparison of the Abbott SARS-CoV-2 IgG II Quant Assay against Commonly Used Canadian SARS-CoV-2 Enzyme Immunoassays in Blood Donor Retention Specimens, April 2020 to March 2021

Kento T. Abe,[a,b] Bhavisha Rathod,[a] Karen Colwill,[b,c] Anne-Claude Gingras,[a,b] Ashleigh Tuite,[d] Ninette F. Robbins,[e] Guillermo Orjuela,[f] Craig Jenkins,[g] Valerie Conrod,[g] Qi-Long Yi,[h,i] ⓘ Sheila F. O'Brien,[h,i] ⓘ Steven J. Drews[j,k]

[a]Lunenfeld-Tanenbaum Research Institute at Mt. Sinai Hospital, Sinai Health, Toronto, Ontario, Canada

[b]Department of Molecular Genetics, University of Toronto, Toronto, Ontario, Canada

[c]Treadwell Therapeutics, Toronto, Ontario, Canada

[d]Dalla Lana School of Public Health, University of Toronto, Toronto, Ontario, Canada

[e]Scientific Affairs, Abbott Transfusion Medicine, Chicago, Illinois, USA

[f]Scientific Affairs, Abbott Transfusion Medicine, Bogotá, Colombia

[g]COVID-19 Serological Screening Laboratory, Canadian Blood Services, Ottawa, Ontario, Canada

[h]Epidemiology and Surveillance, Canadian Blood Services, Ottawa, Ontario, Canada

[i]School of Epidemiology and Public Health, University of Ottawa, Ottawa, Ontario, Canada

[j]Canadian Blood Services, Microbiology, Edmonton, Alberta, Canada

[k]Department of Laboratory Medicine and Pathology, University of Alberta, Edmonton, Alberta, Canada

**ABSTRACT**   Our group has previously used laboratory and commercially developed assays to understand the IgG responses to SARS-CoV-2 antigens, including nucleocapsid (N), spike (S), and receptor binding domain (RBD), in Canadian blood donors. In this current study, we analyzed 17,428 available and previously characterized retention samples collected from April 2020 to March 2021. The analysis compared the characteristics of the Abbott SARS-CoV-2 IgG II Quant assay (Abbott anti-spike [S], Abbott, Chicago, IL) against four other IgG assays. The Abbott anti-S assay has a qualitative threshold of 50 AU/mL. The four comparator assays were the Abbott anti-nucleocapsid (N) assay and three commonly used Canadian in-house IgG enzyme-linked immunosorbent assays (ELISAs) recognizing distinct recombinant viral antigens, full-length spike glycoprotein, glycoprotein RBD, and nucleocapsid. The strongest qualitative relationship was between Sinai RBD and the Abbott anti-S assay (kappa, 0.707; standard error [SE] of kappa, 0.018; 95% confidence interval, 0.671 to 0.743). We then scored each previously characterized specimen as positive when two anti-SARS-COV-2 assays identified anti-SARS-CoV-2 IgG in the specimen. Using this composite reference standard approach, the sensitivity of the Abbott anti-S assay was 95.96% (95% confidence interval [CI], 93.27 to 97.63%). The specificity of the Abbott anti-S assay was 99.35% (95% CI, 99.21 to 99.46%). Our study provides context on the use of commonly used SARS-CoV-2 serologies in Canada and identifies how these assays qualitatively compare to newer commercial assays. Our next steps are to assess how well the Abbott anti-S assays quantitatively detect wild-type and SARS-CoV-2 variants of concern.

**IMPORTANCE** We describe the qualitative test characteristics of the Abbott SARS-CoV-2 IgG II Quant assay against four other anti-SARS-CoV-2 IgG assays commonly used in Canada. Although there is no gold standard for identifying anti-SARS-CoV-2 seropositivity, aggregate standards can be used to assess seropositivity. In this study, we used a specimen bank of previously well-characterized specimens collected between April 2020 and March 2021. The Abbott anti-S assay showed the strongest qualitative relationship

**Ad Hoc Peer Reviewer** Vincent Streva

Address correspondence to Steven J. Drews, steven.drews@blood.ca.

The authors declare a conflict of interest. Steven J. Drews has functioned as a content expert for respiratory viruses for Johnson & Johnson (Janssen), and Anne-Claude Gingras receives research support from Providence Therapeutics Holdings, Inc., for other projects. Guillermo Orjuela and Ninette F. Robbins are current employees and shareholders of Abbott Laboratories.

with a widely used laboratory-developed IgG assay for the SARS-CoV-2 receptor binding domain. Using the composite reference standard approach, we also showed that the Abbott anti-S assay was highly sensitive and specific. As new anti-SARS-CoV-2 assays are developed, it is important to compare their test characteristics against other assays that have been extensively used in prior research.

**KEYWORDS** IgG, SARS-CoV-2 antibody, methods comparisons, nucleocapsid, receptor binding domain, spike

Canadian Blood Services previously engaged a broad group of laboratories in North America to attempt to understand the neutralizing capacity of blood donor antibodies to SARS-CoV-2 (1–5). Originally, much of this preliminary work was focused on supporting SARS-CoV-2 convalescent plasma studies in Canada (1, 2, 4, 5). The identification of waning neutralizing antibody responses in blood donors (1) led to the development of a further "Correlates of Immunity" project, which had the stated goal of understanding changes in anti-SARS-CoV-2-neutralizing capacity as the COVID-19 pandemic advanced. As part of this Correlates of Immunity project, our group was able to sample 1,500 retention specimens a month from Canadian blood donors using a repeated cross-sectional design with random cross-sectional sampling of all available retention samples for a 12-month period from April 2020 until March 2021 (6–10). During this process, we used a variety of assays that have been widely used by our and other groups to assess SARS-CoV-2 seroprevalence in Canada. These included the Abbott Architect antinucleocapsid antigen IgG assay (Abbott-NP, Abbott, Chicago, IL), as well as three in-house Sinai Health (Toronto, ON, Canada) IgG enzyme-linked immunosorbent assays (ELISAs) utilizing recombinant viral antigens, full-length spike glycoprotein (S), spike glycoprotein receptor binding domain (RBD), and nucleocapsid (NP) (2, 3, 6, 11–16). The Sinai Health IgG ELISAs were developed to allow for the scalable parallel detection of IgGs against the S, RBD, and NP. They are described in extensive detail in the literature (12). Apart from work undertaken with Canadian Blood Service, the laboratory-developed automated ELISAs described are being used or have been used in multiple Canadian studies, including the Canadian COVID-19 Antibody and Health Survey from Statistics Canada (17), the Action to Beat Coronavirus study (18), and studies focused on infection and/or vaccine responses across different cohorts predicted to have a weaker immune response (19, 20).

Given the absence of a gold standard (6, 21), we previously characterized SARS-CoV-2 seropositivity, including latent class analysis (7, 8) and composite reference standard approaches (6). We also attempted to understand the impact of donor-declared vaccine history on SARS-CoV-2 serological profiles (1, 2, 9, 10). We have noted that responses of different assays are, at times, inconsistent with one another (1, 6–9). These differences may be due to a variety of factors, including COVID-19 vaccination, false-positive results (probably less likely), assay accuracy and reliability, cross-reactivity with seasonal coronaviruses, and antibody waning for both anti-N and anti-S targets (6–8, 22). In our prior analysis of specimens from the Correlates of Immunity project, we were able to determine that regardless of the approach, by March of 2021, infection and vaccine-mediated seroprevalence in Canadian blood donors ($<$10%) was much lower than U.S. seroprevalence estimates. We also noted that our specimen bank was influenced by relatively low infection rates and the relatively slow ramp-up of vaccination programs (6, 7, 23, 24).

The development of new anti-SARS-CoV-2 commercial assays allows for the operationalization of seroprevalence studies by non-research-focused laboratories, including clinical and public health laboratories as well as blood operators (7, 21, 25, 26). The Abbott SARS-CoV-2 Quant assay (Abbott, Chicago, IL) was developed for the qualitative and quantitative determination of IgG against SARS-CoV-2 S. The qualitative cutoff for this assay has been described as 50 AU/mL (27). This assay has now been used in multiple seroprevalence surveys (28–31). Due to the nonavailability of some retention specimens from our specimen bank, this study did not attempt to infer neutralizing antibody seroprotection from the seroprevalence estimates.

**TABLE 1** Comparison of Abbott-anti-S and Abbott anti-N assays

| Abbott anti-S result | No. (%)[a] of results | | Total |
| --- | --- | --- | --- |
| | Abbott anti-N positive | Abbott anti-N negative | |
| Positive | 134 (28.7) | 333 | 467 |
| Negative | 43 | 16,918 (99.7) | 16,961 |
| Total | 177 | 17,251 | 17,428 |

[a]Numbers in parentheses represent percent agreement versus other methodology.

Instead, we attempted to understand qualitatively the test characteristics of the Abbott anti-S assay against specimens well characterized by qualitative assays previously utilized by Canadian seroprevalence and health studies.

## RESULTS

**Study population characteristics.** Retention specimens from a total of 17,428 blood donors were included in the study, with samples collected between April 2020 and March 2021. Epidemiological characterization of these blood donors was previously described in multiple publications (6, 7).

**Percentage agreement between the Abbott anti-S assays and Abbott anti-N, Sinai anti-S, Sinai anti-RBD, and Sinai anti-N assays.** The percentage agreement estimates between the Abbott anti-S assay and Abbott anti-N, Sinai anti-S, Sinai anti-RBD, and Sinai anti-N assays are listed in Tables 1 to 4. The highest agreement between positive Abbott anti-S assay results was with positive Sinai anti-S results (72.6%; Table 2), then positive Sinai anti-RBD results (66.6%; Table 3), then positive Sinai anti-N results (32.3%; Table 4), and, finally, positive Abbott anti-N results (28.7%; Table 1). The highest agreement between negative Abbott anti-S results was with negative Sinai anti-RBD results (99.5%; Table 3), then negative Sinai anti-N results (97.7%), as well as negative Abbott anti-N results (99.7%; Table 1), and, finally, negative Sinai anti-N results (97.4%; Table 2).

**Comparison of agreement between qualitative results (kappa analysis).** Qualitative determination of positive results used signal-to-cutoff values, which are described in the Materials and Methods. The distribution of qualitative agreement between the Abbott anti-S assays and Abbott anti-N (Table 1), Sinai anti-S (Table 2), Sinai anti-RBD (Table 3), and Sinai anti-N (Table 4) were determined. The highest kappa was with Sinai anti-RBD (kappa, 0.707; SE of kappa, 0.018; 95% confidence interval (CI), 0.671 to 0.743) and progressively lower for Sinai anti-S (kappa, 0.527; SE of kappa, 0.020; 95% CI, 0.489 to 0.565), Abbott anti-N (kappa, 0.407; SE of kappa, 0.030; 95% CI, 0.348 to 0.467), and lowest for Sinai anti-N (kappa, 0.278; SE of kappa, 0.027; 95% CI, 0.226 to 0.3330).

**Analysis of discordant specimens positive by Abbott anti-S.** Of the 467 specimens determined to be positive by the Abbott anti-S qualitative cutoff, distributions of positivity by other assays are identified in Fig. 1 and 2. Discordant specimens positive by Abbott anti-S and negative by all other assays or positive by only one other assay were analyzed as follows.

About a quarter of Abbott anti-S-positive specimens were negative on all other assays (i.e., their signal-to-cutoff values were below cutoff) (Fig. 1). None of these 111 specimens that were only Abbott anti-S positive had a sequentially prior Abbott anti-N-positive specimen (based on Canadian Institutes of Health Research [CIHR] number). None of the 111 specimens were from donors who declared a recent history of COVID-19 vaccination. Not only were target signals below cutoff, but in general, median signal-to-cutoff values were well below cutoff. Summary results for each of the targets suggested low median signal-to-cutoff values in this

**TABLE 2** Comparison of Abbott-anti-S and Sinai anti-S assays

| Abbott anti-S result | No. (%)[a] of results | | Total |
| --- | --- | --- | --- |
| | Sinai anti-S positive | Sinai anti-S negative | |
| Positive | 339 (72.6) | 128 | 467 |
| Negative | 443 | 16,518 (97.4) | 16,961 |
| Total | 782 | 16,646 | 17,428 |

[a]Numbers in parentheses represent percent agreement versus other methodology.

**TABLE 3** Comparison of Abbott anti-S and Sinai anti-RBD assays

| | No. (%)[a] of results | | |
|---|---|---|---|
| **Abbott anti-S result** | **Sinai anti-RBD positive** | **Sinai anti-RBD negative** | **Total** |
| Positive | 311 (66.6) | 156 | 467 |
| Negative | 93 | 16,868 (99.5) | 16,961 |
| Total | 404 | 17,024 | 17,428 |

[a]Numbers in parentheses represent percent agreement versus other methodology.

group of specimens as follows: Abbott anti-N (median, 0.02 [25th to 75 percentiles, 0.02 to 0.05]), Sinai anti-S (median, 0.08 [25th to 75 percentiles, 0.03 to 0.09]), Sinai anti-RBD (median, 0.06 [25th to 75 percentiles, 0.03 to 0.09]), Sinai anti-N (median, 0.06 [25th to 75 percentiles, 0.04 to 0.09]), and Abbott anti-S (median, 78.7 [25th to 75 percentiles, 57.5 to 112.2]).

Of the 467 specimens determined to be positive by the Abbott anti-S qualitative cutoff, about three-quarters were positive on at least one other assay. There were 47 determined distributions of positivity by only one other assay as in Fig. 1. The median Abbott anti-S signal-to-cutoff value for these 47 specimens was relatively low (median, 104.8 AU/mL 25th to 75 percentiles, 72.8 to 117.0 AU/mL). Distributions of other positive assays were Sinai anti-S ($n = 30$), Sinai anti-RBD ($n = 10$), Abbott anti-N ($n = 4$), and Sinai anti-N ($n = 3$). Of the specimens with lone positive assays, the median signal-to-cutoff ratios for positive specimens were generally low for most markers as follows: Abbott anti-N (median, 3.41 [25th to 75 percentiles, 2.09 to 5.05]), Sinai anti-S (median, 0.40 [25th to 75 percentiles, 0.29 to 0.57]), and Sinai anti-RBD (median, 0.23 25th to 75 percentiles, 0.20 to 0.27). Of note, these specimens may have a relatively strong Sinai anti-N (median, 3.34 25th to 75 percentiles, 0.45 to 3.67) signal-to-cutoff values.

**Analysis of discordant specimens negative by Abbott anti-S.** Of the 16,961 specimens determined to be negative by the Abbott anti-S qualitative assay, distributions of positivity by other assays are identified in Fig. 2.

As in Fig. 2, 15 Abbott anti-S negative specimens were determined to be positive by two ($n = 14$, 0.08%) or three ($n = 1$, 0.02%) other tests. Positive values are highlighted in Table 5. Frequencies of positivity were Abbott anti-N ($n = 2$; median, 2.21 [25th to 75 percentiles, 2.05 to 2.37]), Sinai anti-S ($n = 14$; median, 0.32 [25th to 75 percentiles, 0.29 to 0.55]), Sinai anti-RBD ($n = 14$; median, 0.23 [25th to 75 percentiles, 0.22 to 0,44]), and Sinai anti-N ($n = 9$; median, 0.58 [25th to 75 percentiles, 0.53 to 0.72]). Of the 15 specimens, the median signal to cutoff for the Abbott anti-S was relatively low (median, 0.8 [25th to 75 percentiles, 95% CI, 0.2 to 3.2]). In this group, the Abbott anti-S values for two specimens, CIHR013582 (46.8 AU/mL) and CIHR013757 (16.3 AU/mL), were below the qualitative cutoff of 50 AU/mL but above the quantitative reportable limit of detection (6.8 AU/mL) for the Abbott anti-S assay. One specimen (CIHR006065) had positive signals for Sinai anti-S (1.31), Sinai anti-RBD (0.49), and Sinai anti-N (1.92).

**Sensitivity and specificity calculations.** Using a composite reference standard approach, the sensitivity of the Abbott anti-S assay was 95.96% (95% CI, 93.27 to 97.64%). The specificity of the Abbott anti-S assay was 99.35% (95% CI, 99.21 to 99.46%) (Table 6).

## DISCUSSION

This study compared the characteristics of the Abbott SARS-CoV-2 IgG II Quant assay (Abbott anti-spike [S]; Abbott, Chicago IL) against four other SARS-CoV-2 IgG assays that are commonly used in Canada (2, 3, 6, 11–16). This study did not attempt to infer neutralizing

**TABLE 4** Comparison of Abbott-anti-S and Sinai anti-N assays

| | No. (%)[a] of results | | |
|---|---|---|---|
| **Abbott anti-S result** | **Sinai anti-N positive** | **Sinai anti-N negative** | **Total** |
| Positive | 151 (32.3) | 316 | 467 |
| Negative | 392 | 16,569 (97.7) | 16,961 |
| Total | 543 | 16,885 | 17,428 |

[a]Numbers in parentheses represent percent agreement versus other methodology.

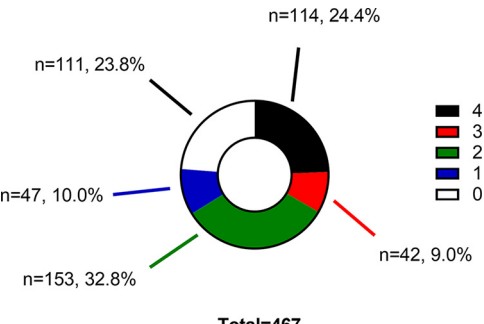

**Total=467**

**FIG 1** Reactivity of Abbott anti-S-positive specimens with other anti-SARS-CoV-2 IgG assays. The graph indicates the percentage and number of Abbott-anti-S-positive specimens that were reactive (1 to 4) and nonreactive by other anti-SARS-CoV-2 IgG assays.

antibody seroprotection from the seroprevalence estimates and did not assess seroprevalence in Canadian blood donors.

The Abbott anti-S assay can be utilized as a qualitative (27, 32), as well as quantitative, assay (33–36) for the detection of anti-SARS-CoV-2 antibodies. The assay is a chemiluminescent microparticle immunoassay for IgG against the RBD region of S (21) with a qualitative assay cutoff of 50 AU/mL (27, 32). In the past, we have also utilized serological assays against N, S, and RBD, including those included in this study, to estimate seroprevalence in Canadian blood donors (6–8). We also noted that well-validated anti-S and anti-RBD assays allow for estimates of seroprevalence that are less impacted by waning antibodies as those seen with anti-N antibodies (6–8, 25, 37). The reasons for waning antibodies to both N and S antigens are still not completely understood but may reflect low levels of infections in some populations, waning immunity after vaccination, immune status in populations studies, and differences in vaccine rollout strategies (38–41). It is also important to utilize a variety of assays for seroprevalence work, given the absence of a gold-standard SARS-CoV-2 immunoassay (6).

Assays that bridge between qualitative binding and quantitative neutralization are important, and this study focused on the qualitative binding element of immunity. In our previous studies, we have shown a difference in detection of binding versus neutralizing anti-SARS-CoV-2 antibodies in specimens from the same cohort of blood donors in our Correlates of Immunity project (9, 10). Other work has shown that not all binding antibodies correlate to neutralization but that IgG against RBD (and to a lesser extent, S) can act as an indicator of neutralization (2, 42). In other cases, detection of both binding and neutralizing antibodies may both indicate immune protection after vaccination in macaque infection models (43). However, even some anti-RBD antibodies may bind to nonneutralizing faces of the RBD molecule (44). We now understand that antibodies to SARS-CoV-2 are polyfunctional and undertake neutralizing and antibody-dependent cell-mediated cytotoxicity (ADCC) and

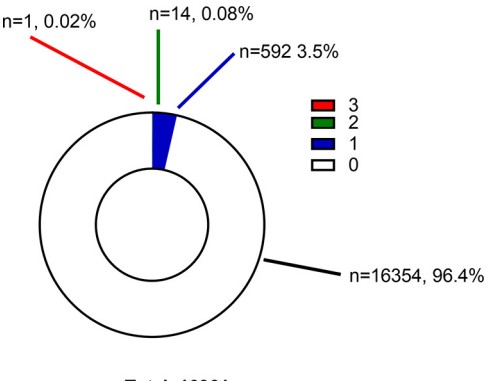

**Total=16961**

**FIG 2** Reactivity of Abbott anti-S-negative specimens with other anti-SARS-CoV-2 IgG assays. The graph indicates the percentage and number of Abbott-anti-S-negative specimens that were reactive (1 to 3) and nonreactive by other anti-SARS-CoV-2 IgG assays.

**TABLE 5** Signal-to-cutoff ratios of four assays on which Abbott anti-S negative specimens were positive by two or more anti-SARS-CoV-2 assays

| CIHR no. | Signal-to-cutoff ratio of: | | | | Abbott anti-S negative specimens (AU/mL) |
| --- | --- | --- | --- | --- | --- |
| | Abbott anti-N | Sinai anti-S | Sinai anti-RBD | Sinai anti-N | |
| CIHR000106 | 0.08 | 0.31 | 0.22 | 0.08 | 3.4 |
| CIHR002075 | 0.05 | 1.00 | 0.43 | 0.63 | 0 |
| CIHR003710 | 0.04 | 0.26 | 0.57 | 0.51 | 0.7 |
| CIHR006065 | 0.02 | 1.31 | 0.49 | 1.92 | 0 |
| CIHR007833 | 0.23 | 0.89 | 0.20 | 0.81 | 0.8 |
| CIHR008235 | 2.05 | 0.29 | 0.06 | 0.23 | 0.2 |
| CIHR009609 | 0.45 | 0.29 | 0.30 | 0.25 | 1.8 |
| CIHR012266 | 0.02 | 0.32 | 0.24 | 0.16 | 3.2 |
| CIHR013582 | 0.02 | 0.44 | 0.60 | 0.02 | 46.8 |
| CIHR013757 | 0.06 | 0.33 | 0.40 | 0.55 | 16.3 |
| CIHR014210 | 2.37 | 0.12 | 0.22 | 0.58 | 0 |
| CIHR015196 | 0.06 | 0.31 | 0.22 | 0.56 | 0.8 |
| CIHR015226 | 0.02 | 0.38 | 0.19 | 0.05 | 1.2 |
| CIHR015837 | 0.01 | 0.30 | 0.23 | 0.61 | 0.8 |
| CIHR016791 | 0.08 | 0.20 | 0.22 | 0.45 | 1.2 |

antibody-dependent cellular phagocytosis (ADCP) through antibodies to both N and S proteins (4, 45–47). However, the mechanisms of ADCC and ADCP are still being understood and require further studies in humans (48, 49).

The characterization of commercially available anti-S quantitative assays, such as the Abbott anti-S assay, is important for supporting large-scale serosurveys and for guiding public health decision-making (25, 37, 50). This ability of the Abbott serology platforms to test both anti-S and anti-N signals will play an important role in helping serosurveillance groups to characterize population-level immune responses to both vaccination and natural SARS-CoV-2 infection (29–31, 51, 52).

We acknowledge several important caveats in this study, including the use of a relatively small number of specimens over a 12-month period from April 2020 to March 2021. Because these were healthy blood donors, these donors did not provide clinical information on COVID-19 disease. The methodologies used to detect antibodies were qualitative or, in the case of the Abbott anti-S assay, were analyzed as qualitative assays and did not assess antibody titers over time. For a small subset of data, we assessed donor-declared vaccine histories and did not access health databases in jurisdictions where donors lived. In this study, we also did not account for the impact of variants of concern on how SARS-CoV-2 immunoassays are qualified and characterized (25, 53).

In conclusion, we describe the qualitative characteristics of the Abbott anti-S assay. We used a composite reference standard approach to estimate the sensitivity of the Abbott anti-S assay to be 95.96% (95% CI, 93.27 to 97.63%). We also estimated the specificity as 99.35% (95% CI, 99.21 to 99.46%). Our study provides context on the use of commonly used SARS-CoV-2 serologies in Canada and identifies how these assays qualitatively compare to newer commercial assays. Our next steps are to assess how well the Abbott anti-S assays quantitatively detect SARS-CoV-2 wild type and variants of concern.

## MATERIALS AND METHODS

**Ethical considerations.** This project received ethics board clearance from the following institutions: Canadian Blood Services, the University of Alberta, and Sinai Health, Toronto (Mount Sinai Hospital).

**TABLE 6** Sensitivity and specificity calculation matrix using reference standards

| Abbott anti-S result | No. of positive specimens (≥2 positive tests) | No. of negative specimens (<2 positive tests) | Total |
| --- | --- | --- | --- |
| Abbott anti-S positive | 356 | 111 | 467 |
| Abbott anti-S negative | 15 | 16,946 | 16,961 |
| Total | 371 | 17,057 | 17,428 |

**CIHR Correlates of Immunity study participants and samples.** Canadian Blood Services has blood collection sites concentrated in large and small cities in all Canadian provinces except Quebec. Blood donors must meet the following criteria: be at least 17 years of age, pass health selection criteria screening, and pass infectious disease screening protocols for blood donations that are then used to manufacture products for transfusion. At each donation, there is also an additional EDTA plasma (Becton Dickson [BD], Mississauga, ON, Canada) retention sample collected for additional blood testing if required (54).

**Study design and population.** We designed a repeated cross-sectional design with a random cross-sectional sampling of all available retention samples ($n$ = 1,500/month) for a 12-month period from April 2020 until March 2021. A two-stage process sampling approach was used with a random selection of blood donor clinics followed by a random sample selection within clinics. Samples were anonymized. We collected variables, including sex, birth year, residential forward sortation area (FSA; first three characters of postal code), donation date, and collection site, which were extracted from the Canadian Blood Services donor database. Retention plasma specimens were aliquoted at Canadian Blood Services and transported to test sites (6). A residual specimen was stored at $-80°C$ for the remainder of the study.

**SARS-CoV-2 antibody testing.** Each retained plasma sample was evaluated for SARS-CoV-2 IgG antibodies using four assays as described previously (6). This study undertook parallel testing for the Abbott Architect anti-nucleocapsid antigen assay (Abbott-NP, Abbott, Chicago, IL), as well as three in-house IgG ELISAs utilizing recombinant viral antigens, full-length spike glycoprotein (spike), spike glycoprotein receptor binding domain (RBD), and nucleocapsid (NP) (11, 12). The testing of specimens by the Abbott-NP assay occurred in a sequential and ascending manner based on specimen number (CIHR number). Residual specimens that were previously tested by the Abbott-NP assay were available for testing by the Abbott SARS-CoV-2 Quant assay. Specimens used for this study included those described in a prior analysis for April 2020 to March 2021, inclusive (6).

**Thresholds used for assay comparisons.** Ratio-converted ELISA reads were undertaken as previously described (2, 11), and cutoffs (positive) for each of the targets were N, ≥0.396; RBD, ≥0.186; and S, ≥0.190 (6). Plasma samples were also tested with the Abbott Architect SARS-CoV-2 IgG test (Abbott Laboratories, USA), which detects anti-N IgG antibodies as directed by the manufacturer, using an antibody index (AI) cutoff of 1.4. Plasma samples were also tested with the Abbott Architect anti-S SARS-CoV-2 IgG test (Abbott Laboratories, USA), which detects the anti-S total with the cutoff of 50 AU/mL.

**Data storage and statistical analysis.** We used a Microsoft Excel (Redmond, WA, USA) spreadsheet for data storage. Data were analyzed as described in the results section using GraphPad Prism (v9.2.0; GraphPad Software, Inc., San Diego, CA, USA). The percentage agreement between the Abbott SARS-CoV-2 Quant assay and other methods was calculated based on the denominators of rows as per Tables 1 to 4 and Table 6. Kappa analysis was performed using GraphPad Prism Quick Calcs and interpretations as previously described (55, 56). Sensitivity and specificity calculations were undertaken using the Vassar Stats clinical calculator 1 with 95% confidence intervals estimated as per Cohen (57, 58).

**Collection of SARS-CoV-2 vaccination history in donors.** All donors at the time of donation were asked if they received a SARS-CoV-2 vaccine in the past 3 months. This was standard practice by Canadian Blood Services. Information on dosing and vaccine type was limited. Provincial vaccine databases are not linked to the blood operator records of donation.

**Determination of anti-SARS-CoV-2-positive and -negative specimens for analysis of sensitivity and specificity.** In the past, we have estimated positives in the sample set using multiple methods, including a composite reference standard where seropositivity using two or more assays described below represents a true-positive case (6). For this study, we continued to use that reference standard. Specimens were deemed to be anti-SARS-CoV-2 positive if they reacted with any two of the following previously characterized assays; Abbott anti-N, Sinai anti-S, Sinai anti-RBD, and Sinai anti-N. Negative specimens were reactive to only one of the assays listed above or none of these assays.

## ACKNOWLEDGMENTS

We thank members of the Gingras laboratory, Adrian Pasculescu, and the Network Biology Collaborative Centre for help with ELISA assays and Yves Durocher at the National Research Council of Canada (NRC) for recombinant antigens used for the ELISA assays. We are also grateful to the Canadian Blood Services staff and leadership for their support of this project.

The following authors have no conflicts of interest: S.F.O.B., A.T., K.T.A., B.R., K.C., and Q.-L.Y. S.J.D. has functioned as a content expert for respiratory viruses for Johnson & Johnson (Janssen), and A.-C.G. receives research support from Providence Therapeutics Holdings, Inc. for other projects. G.O. and N.F.R. are current employees and shareholders of Abbott Laboratories.

Methodology, K.T.A., K.C., C.J., and V.C.; investigation, G.O., N.F.R., B.R., and S.J.D.; funding acquisition, A.T., A.-C.G., S.F.O.B., G.O., N.F.R., and S.J.D.; supervision, A.-C.G., S.J.D., and S.F.O.B.; manuscript drafting, K.T.A. and S.J.D.; data collation and analysis, S.F.O. and Q.-L.Y.; administration, S.J.D.

S.J.D., S.F.O.B., and A.-C.G. received support through the Canadian Institutes of Health Research (CIHR; VR2-172723) and Alberta Innovates (G2020000360 Drews); A.-C.G. also received funding through the Royal Bank of Canada and the Krembil Foundation to the Sinai Health System Foundation, Ontario Together, and CIHR (VR1-172711, with a supplement

from the COVID-19 Immunity Task Force). K.T.A. received funding through an Ontario Graduate Scholarship and a CIHR CGS-D studentship. The robotics equipment used for the ELISA assays is housed in the Network Biology Collaborative Centre at the Lunenfeld-Tanenbaum Research Institute (A.-C.G.), a facility supported by Canada Foundation for Innovation funding, the Ontario Government, and Genome Canada and Ontario Genomics (OGI-139). Commercial Abbott Architect SARS-Cov-2 IgG assay kit costs were partially supported by Abbott Laboratories, Abbott Park, Illinois. Abbott analyzers used at Canadian Blood Services were provided by the COVID-19 Immunity Task Force (CITF). The funders had no role in study design, data collection and analysis, decision to publish, or preparation of the manuscript.

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
