## [Reviewer comments · Microbiology Spectrum]

Microbiology Spectrum

A qualitative comparison of the Abbott SARS-CoV-2 IgG II Quant assay against commonly used Canadian SARS-CoV-2 enzyme immunoassays in blood donor retention specimens, April 2020 to March 2021.

Kento Abe, Bhavisha Rathod, Karen Colwill, Anne-Claude Gingras, Ashleigh Tuite, Ninette Robbins, Guillermo Orjuela, Craig Jenkins, Valerie Conrod, Qi-Long Yi, Sheila O'Brien, and Steven Drews

Corresponding Author(s): Steven Drews, Canadian Blood Services

Review Timeline:

Submission Date:	March 28, 2022
Editorial Decision:	May 2, 2022
Revision Received:	May 10, 2022
Accepted:	May 14, 2022

Editor: Heba Mostafa

Reviewer(s): Disclosure of reviewer identity is with reference to reviewer comments included in decision letter(s). The following individuals involved in review of your submission have agreed to reveal their identity: Vincent Streva (Reviewer #1)

Transaction Report:

DOI: <https://doi.org/10.1128/spectrum.01134-22>

May 2, 2022

Dr. Steven J Drews
Canadian Blood Services
8249 114 St NW
Edmonton, Alberta T6G 2R8
Canada

Re: Spectrum01134-22 (A qualitative comparison of the Abbott SARS-CoV-2 IgG II Quant assay against commonly used Canadian SARS-CoV-2 enzyme immunoassays in blood donor retention specimens, April 2020 to March 2021.)

Dear Dr. Steven J Drews:

Link Not Available

Sincerely,

Heba Mostafa

Journals Department
Reviewer comments:

Reviewer #1 (Comments for the Author):

In this paper, Abe et al. present a modest study of the Abbott SARS-CoV-2 IgG II Quant assay in a moderately sized patient population in Canada, comparing this assay to three other commonly used assays in Canada (one S-based assay, one S (RBD)-based assay, and one N-based assay). The study is largely descriptive, but provides good information that should be published for the field. Some minor comments/suggestions are indicated below:

1. It might be helpful to explain earlier in the manuscript (and in slightly more detail) the information on the Sinai assays (development, use in the field, etc.)

2. Minor typo: line 118 - remove "by" in "...this assay has been described by as 50 AU/mL..."
3. Please check the numbers in Tables 1 and 4. The rows (Abbott anti-S pos and neg) do not always add up to the same number. As per the manuscript text, the rows should add to 467 (Abbott anti-S positive) and 16,961 (Abbott anti-S negative), however in Table 1 they add to 481/16,947 and in Table 4 they add to 468/16,960.
4. I feel the Tables would be more clear if in addition to total numbers, they also contained percentage agreement and/or column/row totals so that a quick comparison and rough sensitivity/specificity comparison could be made.
5. I would be particularly interested in knowing more about the clinical history of the fifteen patients listed in Table 5. Is there anything in the history to help explain the discordant results?
6. Table 6: for consistency, please switch the order of the two columns (Positive first, then negative) so that they match the other tables.

Staff Comments:

Preparing Revision Guidelines

Please return the manuscript within 60 days; if you cannot complete the modification within this time period, please contact me. If you do not wish to modify the manuscript and prefer to submit it to another journal, please notify me of your decision immediately so that the manuscript may be formally withdrawn from consideration by Microbiology Spectrum.

In this paper, Abe et al. present a modest study of the Abbott SARS-CoV-2 IgG II Quant assay in a moderately sized patient population in Canada, comparing this assay to three other commonly used assays in Canada (one S-based assay, one S (RBD)-based assay, and one N-based assay). The study is largely descriptive, but provides good information that should be published for the field. Some minor comments/suggestions are indicated below:

1. It might be helpful to explain earlier in the manuscript (and in slightly more detail) the information on the Sinai assays (development, use in the field, etc.)
2. Minor typo: line 118 – remove “by” in “...this assay has been described by as 50 AU/mL...”
3. Please check the numbers in Tables 1 and 4. The rows (Abbott anti-S pos and neg) do not always add up to the same number. As per the manuscript text, the rows should add to 467 (Abbott anti-S positive) and 16,961 (Abbott anti-S negative), however in Table 1 they add to 481/16,947 and in Table 4 they add to 468/16,960.
4. I feel the Tables would be more clear if in addition to total numbers, they also contained percentage agreement and/or column/row totals so that a quick comparison and rough sensitivity/specificity comparison could be made.
5. I would be particularly interested in knowing more about the clinical history of the fifteen patients listed in Table 5. Is there anything in the history to help explain the discordant results?
6. Table 6: for consistency, please switch the order of the two columns (Positive first, then negative) so that they match the other tables.

To the Editor,

Microbiology Spectrum

Thank you for the reviewer's comments. We will provide a point-by-point response to each of the comments below

1. It might be helpful to explain earlier in the manuscript (and in slightly more detail) the information on the Sinai assays (development, use in the field, etc.)

We have added further descriptions of the Sinai assays in the introductory text.

2. Minor typo: line 118 - remove "by" in "...this assay has been described by as 50 AU/mL..."

This change has been made

3. Please check the numbers in Tables 1 and 4. The rows (Abbott anti-S pos and neg) do not always add up to the same number. As per the manuscript text, the rows should add to 467 (Abbott anti-S positive) and 16,961 (Abbott anti-S negative), however in Table 1 they add to 481/16,947 and in Table 4 they add to 468/16,960.

We have gone through all the data in Tables 1-4 and corrected the Tables. We have also recalculated the Kappa estimates and clarified the method for determination of SE as per Cohen in the references.

4. I feel the Tables would be more clear if in addition to total numbers, they also contained percentage agreement and/or column/row totals so that a quick comparison and rough sensitivity/specificity comparison could be made.

The calculation of a percentage agreements as well as row and column totals were done for Tables 1-4. In Table 6, because this is a sensitivity/specificity matrix table, we show row and column totals.

5. I would be particularly interested in knowing more about the clinical history of the fifteen patients listed in Table 5. Is there anything in the history to help explain the discordant results?

Because these were healthy blood donors, they would not have provided clinical information on COVID-19 disease. We have added this information to the caveats section of the discussion.

6. Table 6: for consistency, please switch the order of the two columns (Positive first, then negative) so that they match the other tables.

Table 6 was modified.

May 14, 2022

Dr. Steven J Drews
Canadian Blood Services
8249 114 St NW
Edmonton, Alberta T6G 2R8
Canada

Re: Spectrum01134-22R1 (A qualitative comparison of the Abbott SARS-CoV-2 IgG II Quant assay against commonly used Canadian SARS-CoV-2 enzyme immunoassays in blood donor retention specimens, April 2020 to March 2021.)

Dear Dr. Steven J Drews:

Your manuscript has been accepted, and I am forwarding it to the ASM Journals Department for publication. You will be notified when your proofs are ready to be viewed.

Sincerely,

Heba Mostafa
Editor, Microbiology Spectrum
